# Dissemination, Implementation, and Evaluation of an Effective School-Based Intervention to Promote Physical Activity in Adolescents: A Study Protocol

**DOI:** 10.3390/bs13040290

**Published:** 2023-03-28

**Authors:** Hisham Bachouri-Muniesa, Léna Lhuisset, Alberto Aibar, Nicolas Fabre, Sonia Asún-Dieste, Julien E. Bois, Maïté Verloigne, José Antonio Julián Clemente, Lionel Dubertrand, José Carlos Ribeiro, Enrique García Bengoechea, Eduardo Ibor-Bernalte, Javier Zaragoza

**Affiliations:** 1Department of Didactics of the Musical, Plastic and Corporal Expression, Faculty of Human Sciences and Education, EFYPAF Research Group, University of Zaragoza, 22003 Huesca, Spain; aibar@unizar.es (A.A.); sonasun@unizar.es (S.A.-D.); jajulian@unizar.es (J.A.J.C.);; 2Laboratory Movement, Equilibre, Performance, Santé (MEPS), Faculty of Sciences and Techniques of Physical and Sports Activities (STAPS), e2s, University of Pau and Pays de l’Adour, 65000 Tarbes, France; lena.lhuisset@univ-pau.fr (L.L.); nicolas.fabre@univ-pau.fr (N.F.); julien.bois@univ-pau.fr (J.E.B.); 3Department of Movement and Sport Sciences, University of Ghent, 9000 Ghent, Belgium; maite.verloigne@ugent.be; 4Maison Sport Santé, City of Tarbes, 65000 Tarbes, France; l.dubertrand@mairie-tarbes.fr; 5Research Center in Physical Activity, Health and Leisure (CIAFEL), Faculty of Sport, University of Porto, 4200-450 Porto, Portugal; jribeiro@fade.up.pt; 6Department of Physical Education and Sport Sciences, Physical Activity for Health Research Cluster, Health Research Institute, University of Limerick, V94 T9PX Limerick, Ireland; 7Research and Evaluation Unit, Sport Ireland, D15 PNON Dublin, Ireland

**Keywords:** school, adolescents, stakeholders, physical activity, replicating effective programs

## Abstract

Adolescents around the world do not engage in sufficient physical activity and the Spanish context is no exception. Understanding the educational context as a complex system, school-based multi-level and multi-component interventions seem to be an effective strategy to reverse this trend. Moreover, a co-creational approach seems to facilitate the mobilization of community partnerships and the engagement of stakeholders in the intervention process. This study aims to describe the dissemination, implementation, and evaluation process of an effective school-based intervention program in another setting using the replicating effective programs framework and a co-participatory approach. This study will be conducted in two Spanish secondary schools located in the region of Aragon (experimental vs. control school) in a sample of adolescents in the second grade (13–14 years old). To evaluate the effectiveness, different health behaviors such as physical activity, sleep, sedentary time with screens, nutrition, and psychosocial variables will be quantitatively measured at baseline and after the implementation of the intervention. Qualitative methods will also be used to better understand the implementation process and the co-creation approach, as well as to provide insights into the sustainability of the intervention program. The current study has the potential to provide strong information about the dissemination, implementation, and evaluation process of school-based programs to promote healthy behaviors among adolescents.

## 1. Introduction

Globally, adolescents’ physical activity (PA) participation rates are quite low, and more than three-quarters (81%) of adolescents do not meet the World Health Organization (WHO) PA guidelines [1]. These percentages of inactivity are currently quite similar in the Spanish context [2]. Considering this worldwide concern, the current WHO ‘Global Action Plan on Physical Activity (2018–2030)’ has stated the need to strengthen the development and implementation of behavioral public health interventions that increase PA opportunities among adolescents [3]. To implement PA promotion programs, schools are considered as one of the most important and relevant contexts to influence adolescents’ behavior [3], not only because they spend the most important part of their weekday time at their facilities, but also because a whole-of-school approach allows for the engagement of a wide range of community members such as teachers, parents, and peers [4], who may play a great influence. Theoretically, this approach should enable the effective tackling of this worldwide problem.

Considering the point of view of school-based program research and the implementation of these programs, it is currently important to analyze the school environment as a complex system [5]. Recently, the adoption of a systemic approach is being tried to solve the problem of physical inactivity [6]. International organizations with public policies and local interventions are making efforts to include this research perspective in their implementation designs [3]. Some reviews [7,8] have found that multi-component interventions are effective initiatives to promote PA among young people, although this evidence seems to not be fully consistent across studies [9]. During the last decade, in the Spanish context, one of those multi-component interventions, which has proven to be effective to promote PA, is the ‘Sigue la Huella’ (‘Follow the Footprint’) intervention program [10], and its subsequent evolution to promote healthy behaviors, called ‘Caminos del Pirineo’ (‘Paths of the Pyrenees’) [11].

In general terms, ‘Sigue la Huella’ was initially a school-based multi-component intervention, lasting for three academic years, based on promising intervention strategies and guidelines to promote PA [4]. Its main aim was to increase adolescents’ daily PA levels as well as to improve their motivational outcomes toward PA [12]. This program evolved to a multiple health behavior intervention program called ‘Caminos del Pirineo’, which lasted one single academic year [11]. Framed in different theoretical frameworks (socioecological model, self-determination theory, and theory of planned behavior), these programs fostered the empowerment of all members of the school community to create a healthier school environment from a broader perspective, in terms of healthy behaviors (e.g., PA, sedentary behaviors [SB], sleep, active commuting, dietary habits, soft drink, tobacco, and alcohol consumption). Basically, the common design of these interventions shared the use of four main characteristics: (i) the social-ecological model which enables the adoption of comprehensive multi-factorial solutions [13] and different psychological theories which permit establishing specific strategies linked to the psychological variables of influence provided by the theories (e.g., self-efficacy); (ii) the co-creational approach, including co-participatory research methods which help to mobilize community partnerships and engage stakeholders across the different phases of the intervention process [14,15]; (iii) the integration of the intervention process in the daily dynamics of the school (e.g., tutorial action, physical education lessons, etc.); and (iv) the empowerment of the faculty involved in the intervention process to increase their leadership [16,17].

Given that these two school-based interventions have shown a great level of effectiveness in improving adolescents’ healthy behaviors [11,12], the need to successfully disseminate this kind of intervention program in other settings becomes of paramount importance, to extensively improve adolescents’ health and reduce the knowledge-implementation gap. However, in fact, there are a minority of effective interventions moving from research into practice [18]. This has usually been seen as a dynamic process that includes dissemination, implementation, and scale-up procedures [19]. Disseminating evidence-based interventions is a complicated task [20], because healthy behavior interventions often include multiple components and different stakeholders, and do not account for varying contextual environments [21]. Moreover, little research exists describing how effective interventions can be disseminated and implemented in other real-world settings by other professionals (i.e., interventions delivered by school employees during their standard practice in the education system) [22]. Consequently, disseminating effective interventions remains a challenge in current society that should be addressed to advance the current knowledge [23]. ‘Sigue la Huella’ is not an exception, and it has not yet been disseminated to other local contexts, thus reaching a greater proportion of the population who could potentially benefit from its application [24].

To guide dissemination and implementation efforts, there are different theoretical frameworks in the literature which can be used [25]. Nevertheless, it should be pointed out that some issues remain regarding these frameworks that may influence their effectiveness. Issues such as the balance required between adequate fidelity to the intervention and accommodating characteristics of organizations need to be reflected upon to reduce the gap between theory, research, and practice [14,15,26]. For instance, using a co-participatory approach in which different stakeholders are actively involved in the development and the implementation of the intervention strategies seems to be a promising approach [27]. From the different existing frameworks, we have considered using the replicating effective programs (REP) framework [28] to guide the implementation and dissemination efforts of this manuscript. The REP framework is a theoretical approach which provides a roadmap for disseminating and implementing evidence-based interventions into different settings, in order to maximize the opportunities for more sustainable interventions. Given that sustainability is one of our major challenges in the evolution of the intervention program, this framework seems to be quite appropriate for our proposal. In addition, and given that the issues mentioned above still remain unsolved, this framework has been applied using a co-participatory approach, too.

The general aim of this study protocol is to describe the dissemination, implementation, and evaluation process of an effective school-based intervention program in other settings using the replicating effective programs framework and a co-participatory approach. Based on the main aim, different specific goals were developed: (1) to describe the methodology and the procedure followed for the dissemination and implementation process of the ‘Sigue la Huella’ intervention program in a new secondary school in another local context; (2a) to assess the effectiveness of the ‘Sigue la Huella’ intervention program after the dissemination in another setting; and (2b) to evaluate the dissemination process of the intervention program in terms of the co-creation process and its future sustainability. This study will be guided by the following question: could we replicate an effective program in a different context by co-creating the intervention with local stakeholders, maintaining its effectiveness and making the intervention sustainable through time?

## 2. Materials and Methods

### 2.1. Study Setting and Participants

This dissemination, implementation, and evaluation process will be carried out at a secondary school in the city of Jaca (Huesca, Aragón, Spain). Jaca is a small-sized city situated in the northern part of the region of Aragon in Spain, and it has approximately 13,500 inhabitants. Two public secondary schools (experimental and control group) will be invited to participate in the project at a meeting co-organized with the regional Department of Education. The eligible population of this study are all second-grade students (13–14 years old), from both secondary schools, who will be informed about this project. The sampling method applied will be purposive and the two schools selected will be divided into an experimental and control school, respectively.

The research sample will finally consist of 75 students (46.7% girls) between 13 and 14 years old; 46 from the experimental school (M = 13.22 years; 47.8% girls) and 29 from the control school (M = 13.14 years; 44.8% girls). All students agreed to voluntarily participate in this project. Written permissions will be obtained from their parents and consent authorizations from the students themselves. All participants involved in the co-participatory groups also signed informed consents.

### 2.2. Overall Study Procedure

Consistent with the literature, the dissemination process followed the REP framework. Moreover, considering the recommendations of previous systematic reviews on school-based PA intervention programs [29], the dissemination will be developed using a co-participatory approach where different stakeholders (e.g., students, teachers, family, and policymakers) were actively involved in the intervention program. The general procedure to conduct this study is presented in Figure 1.

Prior to the pre-test data collection phase, the main stakeholders will be identified, contacted, and selected by the research team and the principal of the experimental secondary school. Based on that selection, two different group structures will be created: a local working group (LWG) and a planning committee (PC). Moreover, the figure of the facilitator emerged as an essential individual role to consider in the implementation process of the intervention program.

The LWG will be a multi-sectoral group made up of representatives of the municipal agencies, more specifically from the main areas related to health, youth, and sports services, as well as two representatives from the research team. The main role of this structure is to give advice about questions related to the intervention program. In addition, the PC is a structure established in the experimental school. The PC included different teachers, one of them being the physical education teacher, the principal, different students, and two representatives from the research team. The principal plays a leading role among the components of this group.

As we have already mentioned, another key characteristic in the development of the co-creation process will be the election and the assignment of a facilitator (i.e., a researcher with a leading role in the intervention process). This facilitator led and coordinated the dissemination, implementation, and evaluation process, fostering a collaborative approach with the working groups of students, teachers, and families during the intervention.

### 2.3. Research Design

#### 2.3.1. Conceptual Framework for the Dissemination and Implementation Process

To guide the dissemination process, we used the REP framework [28]. However, it should be noted that we made some modifications to the standard procedure, in order to better fit the contextual needs of our local context. Table 1 outlines the REP phases, method, and tasks used to disseminate the ‘Sigue la Huella’ intervention program.

As stated by the REP framework, to develop the dissemination process, different regular and interrelated co-participatory approaches will be used: meetings, workshops, ongoing technical assistance, the distribution of an instructional guide, and focus groups.

(a)Meetings: to inform about the dissemination approach, 8 meetings will be conducted (3 in the preparation phase and 5 during the implementation phase). Each meeting took approximately 60 min during the usual school journey. The aims of the meetings are to inform about the main characteristics of the ‘Sigue la Huella’ intervention program, its respective materials, and to guide teachers in the implementation process, as well as to gather information and opinions from the main stakeholders (i.e., LWG and PC). All meetings will be taped, transcribed, and summarized.(b)Workshops: different workshops will be organized to help stakeholders to (i) identify the main problems in the implementation process of health promotion interventions in schools; (ii) understand the dissemination process; (iii) gain insights into evidence-based strategies and tools to enhance health behaviors; and (iv) comprehensively explain the ‘Sigue la Huella’ guide. In addition to formal information, stakeholders incorporated their experience and “real-world” examples along with other intervention strategies.(c)Technical assistance: the research team offered ongoing technical assistance to the PC. This assistance will mainly be carried out by the facilitator and different research team members at a short in-person meeting or by fluent digital contact. Different tasks, where assistance will be required, will consist, for example, of the implementation of REP or procedures related to accelerometer data collection.(d)Instructional guide: in addition to providing ongoing technical assistance, the research team provided a guide with the most important information about the original ‘Sigue la Huella’ intervention program (for further information see https://capas-c.eu/inicio/profesionales/ (accessed on 3 February 2023)). This document allowed stakeholders to understand what we wanted to undertake and to be able to plan the year according to this program.(e)Focus groups: different focus groups were organized with different stakeholders to identify perceptions and beliefs about adolescents’ health behaviors as well as facilitators, and strategies to disseminate, implement, and evaluate the intervention program.

It should be noted that the control school did not receive any information about this co-creation process. Nevertheless, the control school did receive all the material afterward to support future interventions.

#### 2.3.2. Evaluation Process Design

To analyze and better understand the dissemination and implementation procedure followed during the intervention program, an evaluation process, both on the effectiveness of the intervention and on the dissemination process itself, will be planned and carried out throughout the program. The evaluation process will also be co-designed with the PC stakeholders. All instruments will be directly implemented by the participating teachers, with the help of the facilitator and some members of the research team.

A quasi-experimental design was implemented to examine the effect of the ‘Sigue la Huella’ intervention program. Adolescents’ health-related behaviors as well as psychological factors will be assessed at baseline and immediately after the 16-week intervention program with different instruments in both the control and experimental secondary schools. PA was assessed using objective and subjective instruments. The main variables and instruments used for assessing the effectiveness of the intervention program are reported in Table 2.

The actual dissemination and implementation processes will also be evaluated using different dimensions. Detailed information of this process is provided in Table 3.

Co-creation process. To evaluate the co-creation process, qualitative data will be gathered to increase the validity of our findings. Before the implementation and during the post-test intervention, a focus group was conducted with the co-creation groups (LWG and PC) to evaluate the process of being part of these co-creation groups and to evaluate the intervention they themselves had developed. We used a similar interview guide implemented in the original evaluation process of ‘Sigue la Huella’ [39], but this time we included specific aspects of the context, as well as new topics of interest in this program.Sustainability. To value the sustainability of the intervention program, we used a contextualized adaptation of the Program Sustainability Assessment Tool [40]. We implemented this tool in different focus groups at the end of the intervention program. We applied this tool with the PC “leader” (i.e., the principal), specifying the context of the intervention program in each dimension. At the end of the intervention, we also developed different focus groups with the co-creation groups and families to explore their perspectives about the satisfaction with the intervention program and the possibilities of its sustainability in depth.

### 2.4. Data Analysis Plan

In order to analyze the collected data in terms of effectiveness and in relation to the dissemination and implementation process, we will follow a mixed-method confirmatory design [41]. This strategy would be in line with the most recent recommendations for evaluating complex intervention processes, which promote supplementing quantitative studies with in-depth information [42].

#### 2.4.1. Quantitative Data Analysis

Data from the questionnaires and accelerometers will be analyzed using SPSS Statistics (v.25.0). First, descriptive statistics will be obtained from all variables to examine baseline characteristics of the research population and to explore the variable distribution (means, standard deviations, and frequency, as appropriate). The two study groups (control vs. intervention) will be compared using univariate and multivariate analysis of variance (ANOVA and MANOVA). Specifically, these analyses will be used to study PA levels, sedentary behaviors, and sleep outcomes. A significance level of *p* < 0.05 will be applied for all analyses.

#### 2.4.2. Qualitative Data Analysis

Focus groups will be transcribed and entered into NVivo (release 1.7 version) to develop the analysis. Thematic analysis will be used. This method allows to further explore complex and detailed research [43]. We will follow a six-phase model proposed by Braun and Clarke [44]: (1) Three researchers will initially code 10% of the transcripts to be familiarized with the data. (2) This first coding will be deductive based on the focus group protocol. (3) Embracing both issues stated a priori in the protocol guide as well as newly developing themes. (4) Two researchers will review the themes, revise, and update coding schemes in an iterative process. (5) Finally, the codification will be refined to generate specific definitions and names for each of the themes, and (6) to ultimately produce the final report of the analysis.

### 2.5. Ethical Issues

All procedures performed in this study protocol are in accordance with the ethical standards of the 1964 Helsinki Declaration and its later amendments or comparable ethical requirements. The Ethics Committee for Clinical Research of Aragon (CEICA) approved all procedures of this research project in 2021 (Ethics code: PI21/502).

## 3. Discussion

The current study is designed to describe the dissemination, implementation, and evaluation processes of a school-based intervention program aimed at promoting PA, reducing SB, improving sleep as well as motivational and psychosocial outcomes among adolescents in a different context to the original intervention programs. The previous interventions (‘Sigue la Huella’ or ‘Follow the Footprint’ and ‘Caminos del Pirineo’ or ‘Paths of the Pyrenees’) have proven to be effective in a similar context. However, their complexity makes it difficult to apply them to other settings and to make them really sustainable. Actually, these programs are not being applied in the original contexts. This study establishes progress for the initial program by adapting it to a new environment and placing special focus on its sustainability, something that has emerged as a gap between the literature and practice [24,45]. It is well known that educational contexts are complex systems that need to be comprehensively analyzed [46]. For this reason, it is necessary to explore how to systematize the implementation of programs aimed at changing health behaviors in youth.

In order to achieve the aims of the present study (and also to maintain a certain level of fidelity with the original programs), we have designed this intervention from a co-creational approach and by means of a co-participatory research process [14,15] using the REP framework [28]. We wanted to conduct the intervention from a multi-component perspective, since it seems to be an efficient strategy for the effectiveness of these programs [7,8]. A structured protocol has been conceived to develop a program that influences different variables related to health (focusing especially on PA) to give a comprehensive view of the intervention [47]. In addition, another key characteristic of this intervention is the involvement of all stakeholders within the adolescents’ environment in order to bring about more consistent changes with respect to their behavior [47].

Considering the evaluation process, we strongly believe that the quantitative and qualitative data collected should lead to a better understanding of our research goals [29]. We will apply a confirmatory mixed-method design to perform the data analysis as this has been used in other similar interventions [48,49,50]. Moreover, this study should also provide some insights into the possible sustainability of this type of school-based program, which integrates the context, culture, and population of the intervention into the design of the daily dynamics of the school [51].

This study may have some limitations due to the reduced sample of adolescents participating in the intervention program or the use of self-reported data in the evaluation process. Nevertheless, the use of quantitative and qualitative methods constitutes a strength of paramount importance for this study.

## 4. Conclusions

In conclusion, the present study should highlight the key points to disseminate, adapt, and evaluate an effective school-based intervention program in other different school contexts. Apart from the effectiveness of the intervention, this study should also provide insight into the difficulties or the opportunities to create sustainable school-based programs. To sum up, our study has the potential to provide valuable information about the dissemination of programs aimed at promoting healthy behaviors. All this information may be used by future researchers, health professionals, policymakers, and the school faculty in order to help them to transfer effective programs to other real-world settings and make them more sustainable.

## Figures and Tables

**Figure 1 behavsci-13-00290-f001:**
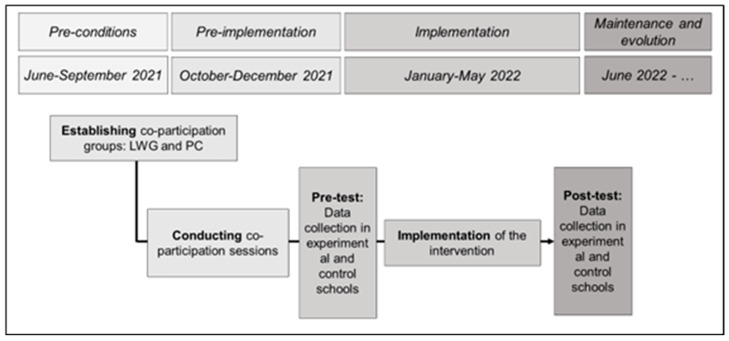
Overall study procedure.

**Table 1 behavsci-13-00290-t001:** Replicating effective programs: phases, co-participation activities, and tasks.

Phase	Co-Participation Activities	Tasks
Pre-conditions	Meetings Focus groups	First contact with the principal and the staff of the city council.
		Present the ‘Sigue la Huella’ intervention program to the school’s educational community. Define the co-creation with stakeholders and provide them with a detailed and adapted document.
		Establish the local working group (LWG) in the school and planning committee (PC) from the city and organize their participation and responsibilities (leadership and coordination).
		Assign a facilitator to the experimental group and specify his/her functions.
Pre-implementation	Meetings Distribution of an instructional guide Workshops	Identify barriers, facilitators, and strategies to disseminate, adapt, implement, and evaluate the program.
		Review and adapt the intervention program, materials, and action plan to the specific context. Define specific intervention strategies.
		Contact with other interesting stakeholders for the intervention (i.e., families, sports clubs, professionals).
		Identify training needs and establish a training plan for PE and tutoring teachers.
		Coordinate and connect the LWG and PC.
		Perform diagnostic evaluation (quantitative) and analyze the data to adapt the program.
Implementation	Meetings Ongoing technical assistance	Apply the adapted intervention program in the experimental group.
		Hold coordination meetings between the LWG, PC, and the facilitator.
Maintenance and evolution	Ongoing technical assistance Focus groups	Evaluate the program at the end.
		Communicate results to all stakeholders.
		Assess stakeholders’ satisfaction with the project and their perceptions of its potential sustainability.

**Table 2 behavsci-13-00290-t002:** Variables and instruments to evaluate the effectiveness of the intervention program.

Variable	Instrument	Dimension	Reference
Physical activity and sedentary time (objectively)	ActiGraph GT3X+ accelerometer	A total of 10 h of movement registered per day for at least 3 days during 1 week and 1 day at a weekend	Cut-points of [30]
Physical activity and sedentary time (subjectively)	Youth Activity Profile—Spain (YAP-S)	Physical activity in and out of school during the last week	[31]
Future intention to practice physical activity	Three ad-hoc questions	Intention to practice physical activity in the following five weeks, from theory of planned behavior questionnaire	[32]
Sleep	Pittsburgh sleep quality index	Bedtime and wake-up time	[33]
Sedentary screen time	Screen time-based behavior questionnaire	Customary time devoted to several sedentary screen-time behaviors during both week and weekend days	[34]
Healthy and unhealthy nutrition	WHO Health Behavior in School Children Survey (HBSC)	Frequency of eating different healthy and unhealthy food	[35]
Self-concept	AF5. Self-concept form 5	Dimensions: Academic; Social; Emotional; Family; Physical.	[36]
Perceived autonomy support from peers, teachers, and family	Perceived Autonomy Support Scale in Exercise Settings (PASSES)	Perceived autonomy support for exercise settings from different stakeholders	[37]
Motivation to exercise	Behavioral Regulation in Exercise Questionnaire—3 (BREQ-3)	Related to self-determined theory, this questionnaire assesses external, introjected, identified, intrinsic, and amotivated forms of regulation for exercise behavior	[38]

**Table 3 behavsci-13-00290-t003:** Dimensions and instruments to evaluate the dissemination and implementation processes.

Dimensions	Instrument	Resources	Assessment Time
Co-creation process	Focus group with different stakeholders	Following a similar protocol used in the evaluation process of ‘Sigue la Huella’ [39]	During the intervention process
Sustainability	Adaptation of Program Sustainability Assessment Tool (PSAT) Focus group with different stakeholders	[40]	Post-intervention

## Data Availability

Data may be made available at a reasonable request by contacting the corresponding author.

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
