# Peer review of "Dissemination, Implementation, and Evaluation of an Effective School-Based Intervention to Promote Physical Activity in Adolescents: A Study Protocol"

_behavsci, 2023, doi:10.3390/bs13040290_

Round 1

Reviewer 1 Report

This study protocol describes the dissemination, implementation and evaluation process of an effective school-based intervention programme in another setting using the REP framework and co-participatory approach. The manuscript is well-written and follows a logical order. My only concern is that the protocol does not provide details on the data analysis plan for both quantitative and qualitative study design. So, it should include a statistical analysis section and a detailed analysis plan.

Author Response

RESPONSE TO REVIEW COMMENTS

Reviewer #1: This study protocol describes the dissemination, implementation and evaluation process of an effective school-based intervention programme in another setting using the REP framework and co-participatory approach. The manuscript is well-written and follows a logical order. My only concern is that the protocol does not provide details on the data analysis plan for both quantitative and qualitative study design. So, it should include a statistical analysis section and a detailed analysis plan.

Answer: We truly appreciate your kind words regarding the efforts made to write this study. Likewise, we would like to thank you for the comments made in

your review to try to improve the quality of our manuscript.

We have incorporated a “2.5. Data analysis plan” section at the end of the Material and Methods section (start at line 244).

Reviewer 2 Report

This research project attempts to investigate the effects of school-based intervention program, Sigue la Huella, on secondary school students’ physical activity (PA) levels and their motivation towards PA in Spain. The research project looks prospective and will provide several pedagogical implications for physical education teachers. There are still some issues to be addressed prior to the acceptance. I will list my point-by-point comments as follows.

1. It is suggested that the research questions should be proposed.

2. It is suggested that the authors should point out what sampling method were adopted.

3. There are several questionnaires adopted to evaluate the effectiveness of the intervention program. The validity and the reliability in previous studies should be reported.

4. A section of data analysis should be added to help readers know how the data will be analyzed.

5. The author(s) adopted quantitative and qualitative methods to evaluate the intervention program. It is suggested that they should emphasize the importance of mixed methods research. In addition, which types of mixed methods designs do they plan to adopt? Convergent mixed methods design, explanatory sequential mixed methods design or exploratory sequential mixed methods design?

Author Response

RESPONS TO REVIEW COMMENTS

Reviewer #2: His research project attempts to investigate the effects of school-based intervention program, Sigue la Huella, on secondary school students’ physical activity (PA) levels and their motivation towards PA in Spain. The research project looks prospective and will provide several pedagogical implications for physical education teachers. There are still some issues to be addressed prior to the acceptance. I will list my point-by-point comments as follows.

Answer: We would like to thank you for the quality and depth of the revision carried out that helped us to improve the quality of our manuscript greatly.

We hope to have resolved these theoretical and methodological issues in the final version of the manuscript. Below, you will find the answers to each specific comment point-by-point that you have made on our manuscript.

  1. It is suggested that the research questions should be proposed.

Answer: We have included a new research question at the end of the introduction section (line 122):

“Could we replicate an effective program in a different context by co-creating the intervention with local stakeholders maintaining its effectiveness and making the intervention sustainable along time?”

  1. It is suggested that the authors should point out what sampling method were adopted.

Answer: We have included the sampling method into the “2.1. Study setting and participants” section (line 134):

“The sample method applied will be purposive and the two schools selected will be divided in experimental and control school respectively.”   

  1. There are several questionnaires adopted to evaluate the effectiveness of the intervention program. The validity and the reliability in previous studies should be reported.

Answer: We have included changes in Table 2 incorporating validation references of the different questionnaires that you commented on. Thanks for your recommendation.

  1. A section of data analysis should be added to help readers know how the data will be analyzed.

Answer: We have included a “2.5. Data analysis plan” section where we have tried to incorporate a detailed description of all the analysis procedures that we are going to carry out in the study.

  1. The author(s) adopted quantitative and qualitative methods to evaluate the intervention program. It is suggested that they should emphasize the importance of mixed methods research. In addition, which types of mixed methods designs do they plan to adopt? Convergent mixed methods design, explanatory sequential mixed methods design or exploratory sequential mixed methods design?

Answer: We have included, also in the “2.5. Data analysis plan”, an explanation of what you comment here, in the first paragraph, we have made a description of the procedures that we will use in the analysis phase. Specifically, we will use a mixed-method confirmatory design.

Reviewer 3 Report

After reading the text submitted for evaluation, it was possible to verify the following issues:

 Both the title and the abstract and the keywords reflect the contents of the text. References up to the year 2022 are reflected in the bibliography.

The work presents a correct scientific structure, which allows a simple and structured reading of it.

The article explains in detail the processes for the implementation of the educational interventions carried out, detailing their characteristics, schools and participants. Throughout the work, the characteristics of the study implemented in schools are mentioned in detail.

The article mentions that a quantitative data collection methodology has been carried out. The data collection instruments are mentioned, but in the article there is not a detailed analysis of all these data. Only the methodology, the participating sample, and the characteristics of the project are mentioned. The text submitted for evaluation lacks data analysis. It is a descriptive work.

It is recommended, either, to change the focus of the work, mentioning that it is a descriptive work of the characteristics of an intervention carried out, or to carry out a detailed analysis of the data that has been mentioned as having been collected. In the second recommendation, there must be an in-depth discussion with the scientific literature

Author Response

RESPONSE TO REVIEW COMMENTS

Reviewer #3: After reading the text submitted for evaluation, it was possible to verify the following issues:

Both the title and the abstract and the keywords reflect the contents of the text. References up to the year 2022 are reflected in the bibliography.

The work presents a correct scientific structure, which allows a simple and structured reading of it.

The article explains in detail the processes for the implementation of the educational interventions carried out, detailing their characteristics, schools and participants. Throughout the work, the characteristics of the study implemented in schools are mentioned in detail.

The article mentions that a quantitative data collection methodology has been carried out. The data collection instruments are mentioned, but in the article there is not a detailed analysis of all these data. Only the methodology, the participating sample, and the characteristics of the project are mentioned. The text submitted for evaluation lacks data analysis. It is a descriptive work.

It is recommended, either, to change the focus of the work, mentioning that it is a descriptive work of the characteristics of an intervention carried out, or to carry out a detailed analysis of the data that has been mentioned as having been collected. In the second recommendation, there must be an in-depth discussion with the scientific literature.

Answer (data analysis): Thank you for your suggestion. We really think that your comment has helped us to improve the paper. We have included a “2.5. Data analysis plan” section (starts at line 244) because there were several mistakes in the tenses. It is not a study carried out yet, therefore the time used should have been the future. We have made these changes and improved the language with a native English colleague. Thank you very much for your recommendation:

Answer (discussion section): Thank you for your suggestion. Initially, this section was developed more broadly, without getting too much into a deep discussion of the literature. However, we have modified the discussion by including more depth in the foreseeable findings according to the literature on the subject. We have included the related paragraphs in lines 285, 294 and 300.
